# Design and Performance of Layered Heterostructure Composite Material System for Protective Armors

**DOI:** 10.3390/ma16145169

**Published:** 2023-07-22

**Authors:** Farah Siddique, Fuguo Li, Mirza Zahid Hussain, Qian Zhao, Qinghua Li

**Affiliations:** 1State Key Laboratory of Solidification Processing, School of Materials Science and Engineering, Northwestern Polytechnical University, Xi’an 710072, China; farahsiddique@mail.nwpu.edu.cn (F.S.); mirzazahidhussain@mail.nwpu.edu.cn (M.Z.H.); zq2019@mail.nwpu.edu.cn (Q.Z.); qinghual@nwpu.edu.cn (Q.L.); 2National Innovation Center of Forging and Ring Rolling Technology in Defense Industry, Northwestern Polytechnical University, Xi’an 710072, China

**Keywords:** layered heterostructure composites, energy absorption, elastic collision, armor protection, numerical simulation

## Abstract

A new layered heterostructure composite material system (TC4 as front layer and 2024Al alloy as back layer) was developed and analyzed for its design and performance in terms of an enhanced absorption capability and anti-penetration behavior. The Florence model for energy absorption was modified, so that it can be utilized for the layered heterostructure composite material system with more efficacy. Numerical simulation through Ls-Dyna validated the analytical model findings regarding the energy absorption of the system and both were in good agreement. Results showed that two ductile materials with diverse properties, the hardness gradient and varied layer thickness joined together, specifically behaved like a unified structure and exhibited elastic collision after slight bending, which is possibly due to the decreased yield strength of the front layer and increased yield strength of the second layer. To validate the analytical and numerical findings, the samples of the layered heterostructure composite material system were subjected to a SHPB (Split Hopkinson pressure bar) compression test. The deformation behavior was analyzed in the context of the strain energy density and stain rate sensitivity parameter at different strain rates. The encouraging results proposed that two ductile materials with a hardness gradient can be used as an alternate structure instead of a brittle–ductile combination in a layered structure.

## 1. Introduction

Layered heterostructure composites have proven themselves as one of the most competent material systems from an application perspective, as they potentially enhance the mechanical properties of monolithic material systems. Therefore, they find wide applications in aerospace, armor systems and various other industrial sectors. The combination of two different materials via different processing routes enables them to gather favorable properties together and utilize them as per a service requirement. Although the processing routes have always been the main point of concern, a lot of research has narrowed down the suitable manufacturing techniques for such heterostructure composite material systems [1,2]. As these structures comprise alternate hard and soft layers, when subjected to impact, they behave as a unified structure and there is an even distribution of stress and strain. This, in turn, improves the uniform deformation capability of a structure as well as its combined deformation compatibility. The most commonly layered composite structures used during recent years comprise ceramic–ceramic, metal–ceramic, metal–metal, etc. The interface/boundary between these layers plays a critical role in determining their combined mechanical properties and collective response towards impact, and the most desirable properties are toughness and ductility (soft layers) combined with strength (hard layers) [3].

The most adoptive way to design a composite structure is by selecting appropriate materials as per the service requirement of protective armor. In addition, it is also believed that the fracture toughness, fatigue crack growth and impact resistance of composite structures can be enhanced through both an extrinsic and intrinsic mechanism. The intrinsic properties can be improved by controlling the microstructural parameters like the grain size, orientation, inter-grain distance, etc., and increasing resistance to crack initiation and propagation. However, the extrinsic mechanism improves the aforementioned properties by minimizing the driving forces that can facilitate the crack growth, such as crack bridging, crack deflection and trapping. The extrinsic mechanism involves the effect of the thickness of the layer, interface strength, properties of layers and residual stress, etc. Layered/laminated metal composites adopt the extrinsic mechanism to improve the impact resistance and other allied properties. Hence, these structures can possess very desirable properties for lightweight applications demanding a high strength, stiffness and fracture toughness [4,5,6,7]. The heterostructure composite layered structure is being opted for extensively as protective armors in the defense industry due to its lightweight and superior mechanical properties. These composite structures can best be designed by understanding the deformation and damage mechanisms involved when subjected to high-speed impacts [8]. The progression of material development in correlation with the mechanics of these newly developed material systems has made it equitably possible to devise a potential solution for improvement in protective armor systems. In order to optimize the protective armor material system, the two most influential parameters are as follows. The first one is the development of new materials (more effective than existing materials), and the second one is the design of a protective armor system comprising such an arrangement of materials that enhances the corresponding protective properties. Foregoing in view, the functionally gradient composites possessing inhomogeneous properties along their thickness direction make them a most prevailing entrant for protective armor systems. Many researchers have proposed multiple ways and means through mathematical modeling to analyze the anti-penetration capability of metallic targets as well as multilayer metal targets, combined or spaced, etc. [9,10,11].

Single-layered armor materials generally have a high areal density. Therefore, the single layer may be replaced by multilayers composed of different lightweight alloys such that, in the combined form, they possess less areal density. The resultant composite structure must be lightweight and should possess a high strength that can be utilized for protective armor. In such a case when a projectile hits a target, its first encounter is with the front layer (usually ceramic/composite material, etc.). The incoming projectile travels quickly through the front layer and either perforates it or is self-destroyed in the case of hard brittle material, while it gets eroded and faces ballistic resistance through the bending/petalling of the plate in the case of the ductile front layer. In the case of the front ceramic layer, the projectile enlarges the perforation area and becomes conical in shape. In the second stage, if the projectile succeeds in reaching the composite plate and has enough energy, it will compress the backing plate, which in turn absorbs the remaining amount of energy and deforms accordingly. During the third stage, the projectile comes into direct contact with the composite and the contact area is larger than the one made previously during the projectile–front-layer encounter. The wave front will preferably move in the transversal direction and much less movement will be along the circumferential direction. The main focus of research has been on the front hard layer, preferably ceramics backed up by a ductile layer.

Among various developed models to explain the perforation of a two-layered ceramic structure, the Florence model has found wide acceptance among researchers and is mostly used to optimize armor systems. This basic model has been investigated and improved progressively over time and all of this has considered the areal density and thickness of an armor system as fundamental parameters [12,13,14]. The Florence model [15] focuses on energy absorption, its balance and the ballistic speed limit. Further study led to the Woodward model [16], which was based on a penetration analysis in terms of residual mass and impact velocity and, similarly, many other studies have also been carried out for the design optimization of armor systems based on the Florence model in correlation with numerical simulations and experimental results [17,18]. All the investigations conclude that the material deformation behavior during impact (low and high) is the fundamental mechanism that needs to be well understood. This may be achieved through experiments as well as constitutive models by encompassing morphological aspects and mechanical properties.

In addition, computer simulations can best optimize the combination of materials in composite structures by carrying out simulations for projectile penetration in the proposed design structures and analyze system behavior upon high-speed impacts [19]. The most optimized structure can be manufactured to perform ballistic tests and further promising enhancements in the design concept can be conducted if required. Three important tools are as follows: constitutive models to describe the material response in terms of deformation and failure over a wide range of strains, strain rates and temperatures; computational/numerical models to translate these deformation behaviors by selecting an appropriate material model and experimental methods to obtain the material input parameters for constitutive models; as well as computational codes for simulations that are collectively required to improve the simulation fidelity that can generate better solutions. Therefore, the present study was initiated to modify the original Florence model by combining it with the Thomson model, mainly used for a monolithic layer, in order to elaborate on the energy absorption mechanism in a layered heterostructure composite material system. The proposed model can potentially be used to design a composite structure system comprising different alloys as an armor system against an ogival-nose-impacted body. The bilayer composite structure (TC4 as front layer and 2024Al alloy as back layer) has been proposed to reduce the weight up to almost 19% against armor steel under the same operating conditions without compromising its anti-penetrating capability. The same has been validated through numerical simulation by using Ls-Dyna code. Both analytical and numerical results were found to be in good agreement and further validated with SHPB compression tests and the results are quite encouraging. The main aim is to join easily available alloys such that the best combination of mechanical properties can be achieved and the cost involved in new alloy system development as well as manufacturing difficulties may be avoided. The benefit of the current study is the combination of extensively used aerospace/aircraft materials in the proposed composite structure, which make it expedient from the perspective of manufacturing cost and new material development cost.

## 2. Materials and Experimental Methods

In the present study, TC4 was used as the front layer confronting the incoming threat due to its high specific strength, excellent fatigue resistance, good heat resistance, stable structure and low density. Its density is 4.42 $g/{\mathrm{cm}}^{3}$, yield strength is 1083 MPa and elongation % is ≥13, as reported in the literature. For these outstanding properties, it is widely used in the aerospace and aircraft industry. Being an effective replacement for steels, the utility of TC4 has increased manifold since its invention [20,21]. The backup layer was selected as the 2024Al alloy, as it is the Al-Cu alloy with an excellent strength to weight ratio and good fatigue resistance. It has a density of 2.7 $g/{\mathrm{cm}}^{3}$, Young’s modulus of 73 GPa and yield strength of 280 MPa. Due to its significant mechanical properties, this alloy is used in high-load parts and components of aircrafts. It is therefore proposed to combine these two superior alloys to obtain an enhanced protective armor system with less weight and the combination of excellent mechanical properties. Square-shaped specimens [22,23] with dimensions of 5 mm were machined through an EDM machine.

The significant difference in chemical and thermos-physical properties of TC4 and the 2024Al alloy, for example, the crystal structure, linear coefficient of expansion, thermal conductivity, melting point, etc., results in the formation of intermetallics during joining processes, which are considered to be a big hinderance in achieving excellent mechanical properties of the two alloys [24,25,26]. As mentioned earlier, the interfacial layers potentially generate intermetallics, which are known to weaken the joints; therefore, the morphology of the interfacial layer is of vital importance. Foregoing in view of the above and the structural requirements of the proposed heterostructure composite material system, the formation of intermetallics is not desirable. Hence, thin samples from TC4 and the 2024Al alloy were cut as per specified dimensions and were joined together by using a commercially available chemical adhesive. This arrangement avoided the formation of intermetallics and their adverse effect on the joint performance of the structure. Future-oriented industrial applications are expected to adopt explosive welding methods. Therefore, the two layers were bonded through the chemical adhesive so that the effect of intermetallic formation via other routes could be avoided. The thickness for the upper/front layer was scaled down from the proposed value and adjusted as 2 mm while the lower/back layer was 2.8 mm in the layered structural arrangement, as shown in Figure 1.

The proposed heterostructure composite material system was subjected to a Split Hopkinson pressure bar (SHPB) test. The SHPB apparatus is shown in Figure 2. The samples comprising the upper/front layer of TC4 with the lower/back layer of the 2024Al alloy were prepared for the SHPB test. Three samples were tested under a quasistatic condition (0.01/s) at room temperature on an Instron 3382 universal testing machine as per the ASTM standard, while three samples each were tested under dynamic conditions, 10^3^/s and 6.5 × 10^3^/s, at room temperature on SHPB equipment. The test schematic is shown in Table 1.

## 3. Results and Discussion

The average obtained result of the quasistatic test is shown in the stress–strain curve (Figure 3). It can be seen that at a 0.01/s strain rate and room temperature condition, the stress–strain curve [27] shows an initial increase in the stress that goes abruptly to approximately 300 MPa; after that there is a slow increase in stress that goes up to 455 MPa. At a 0.45 strain and onwards, there is again an abrupt rise, of approximately 730 MPa, in the stress value. As a result of gradual loading during the quasistatic test, the front/upper layer of TC4 acted as a tool and transferred the load to the 2024Al alloy. The long loading time resulted in the failure of the second/lower layer of 2024Al and the load was potentially transmitted back to TC4 just like a spring back action, which resulted in the slopy increase in stress after a flat portion in Figure 3. Resultantly, TC4 remained unaffected but the 2024Al alloy was ruptured.

From Figure 3, it is obvious that the stress–strain curve is divided into two regions: an elastic region and plastic region. The elastic region goes up to the yield point and here it is segregated into two parts. The first part is until the elastic limit where there is an abrupt increase in stress at less strain and the second part extends up to the yield point where, at maximum stress under incessantly increasing strain, the material recovers its initial state. The first part of the stress–strain curve exhibits almost a linear relationship, thereby following Hooke’s law, and its slope gives the elastic modulus of the structure, which is approximately 10.7 GPa; this value indicates the initial deformation behavior is of TC4 only as the wave is still in the propagation mode moving in the forward direction towards the backing layer. While entering the second part, the slope of the curve starts declining until the yield point is reached at a 0.2 strain; here, the slope of the curve approaches zero and the stress at the yield point shows the yield strength of the material, which is approximately 455 MPa (lower than yield strength of TC4 and higher than yield strength of 2024Al alloy). It shows that the extension of the elastic limit to the yield point might involve the combined deformation behavior of the layered composite structure (TC4 + 2024Al alloy) in which the elastic limit of the front layer material is correlated with the elastic region of the second layer. After this point, the plastic deformation starts and involves both strain softening and strain hardening, as is obvious in Figure 3. Strain softening leading to strain hardening might be a result of change in the micro structure (equiaxed fine grains) across the layered composite material after which there is an abrupt rise in stress, leading to material rupture. This deformation pattern shows the increased load bearing capability and enhanced ductility [28].

Resultantly, the flat portion in the stress–strain curve corresponds to the load bearing capacity of the 2024Al alloy. The stress–strain curve here shows the occurrence of three consecutive deformation mechanisms including elastic deformation, plastic deformation and failure/fracture. At a low speed, initially, the elastic deformation occurs in the front layer of TC4 and the material obeys Hooke’s law, which is given by
(1)F=Kx
where K is a constant and shows the stiffness of material, while *x* is the deformation extent in terms of displacement.

The dynamic compression test was carried out on a Split Hopkinson pressure bar at room temperature and 10^3^/s and 6.5 × 10^3^/s strain rates simultaneously, and stress–strain curves are shown in Figure 4. At a 1000/s strain rate (velocity of striker bar is 7.0 m/s), the deformation mode was elastic in both the layers, and due to an uneven stress distribution, the stress–strain curve shows wave-like behavior [29,30]. There is continual change in stress, but due to a high speed of impact, not enough time is available for the deformation and damage to occur.

During the dynamic deformation processes, Equation (1) can be attached in terms of the simple Bachofen formula [31] for describing the stress–strain relationship and is given by
(2)$$\sigma =k{\dot{\varepsilon }}^{m}$$

Here, σ is the flow stress, k is a constant (strengthening) that depends upon the material fundamental properties and can be obtained through experimental data, $\dot{\varepsilon }\;$ is the strain rate and m is the strain rate sensitivity index.

The value of *m* can be obtained by using the following equation:(3)m=lnσσrlnε˙ε˙r 

In the above equation $\sigma_{r}$ and ${\dot{\varepsilon }}_{r}$ are the flow stress and strain rate, respectively, taken as a reference for a quasistatic test. The value of m is calculated as 0.88 at a 1000/s strain rate and 0.70 at 6500/s, as can be seen from Figure 5a. Furthermore, it is also obvious from Figure 5b that at a higher strain rate, the value of *m* surprisingly decreased, which might be the cause of sample failure at a much higher strain rate.

Here, the deformation gradient can be associated with the strain energy density that determines the yielding of material under given stress conditions. It also translates the elastic behavior of a material. Therefore, Equation (2) can be re-written in terms of the strain rate dependency on the strain energy density [32] as follows:(4)$$\hat{U}=k_{\hat{u}}{\dot{\varepsilon }}^{m}$$

Here, $\hat{U}$ represents the strain energy density, and $k_{\hat{u}}$ is the material parameter that can be obtained through experimental data. The strain energy density along with corresponding stress were plotted as shown in Figure 6. It can be seen that at a 1000/s strain rate, there is a negligible increase in the strain energy density at low strain values except a 0.02 strain (orange line) and an increasing trend in the strain energy density is then shown, which reveals a better load-bearing capacity at this specific strain rate with a higher m value; however, when the strain rate is increased to 6500/s, the oscillating behavior in the strain energy density is similar but with a lower value of m. This can possibly be the cause of fracture in the sample at a 6500/s strain rate.

It can be inferred that at a much higher strain rate, the elastic wave is transformed into a plastic wave by the time it reaches the back layer of 2024Al. The forward stress is produced in the front layer, which is transmitted to the back layer as back stress, and as the back stress becomes dominant, it may cause the back layer to fail/rupture instead of getting strong. Here, a significant plastic deformation took place and the 2024Al alloy ruptured, yet due to a sufficient amount of time, the stress wave is reflected back towards the front layer of TC4, thereby raising the secondary increase in stress to its peak value. This also shows that the increased strain rate reduced the strain energy density and, in turn, the load bearing capacity of the layered composite structure, and it became fractured. In this case, the only active and influential material property is the comparatively high stiffness of the front layer compared to the back layer. As a result, there was a maximum force shifting towards the back layer and it resulted in the subsequent failure of the back layer only.

At a high velocity, the deformation behavior changed and the elastic wave was followed by a plastic wave. Therefore, the behavior can be hypothetically summarized as
(5)$$\sigma =k{\dot{\varepsilon }}^{m}+\bar{m}v$$

In the above equation, the first term indicates the deformation behavior that corresponds to the elastic wave whereas the second term translates the continual plastic wave (mass of second layer, $\bar{m}$ and directional stress wave velocity, v). Here, it can be inferred that the fracture of the sample occurred due to the intense plastic wave represented by $\bar{m}v$, which also implies that the deformation behavior was also influenced by Young’s modulus *E* as well as the size of the sample. Under the combined effect, the layered composite structure acts as hard material with a better impact resistance by following the hetero-deformation-induced (HDI) hardening mechanism. Due to less response time, it is believed that during the high-velocity impact, the two different materials acted in a unified form and the back layer was strengthened under the cover of the front layer. It can be assumed on this basis that the layered composite structure can potentially behave as a strong structure with an enhanced anti-penetration capability. As the impact resistance is inversely proportional to Young’s modulus, this is why materials with a high stiffness show less impact resistance and ones with a high toughness show a good impact resistance. The combined effect of these varied properties can be beneficial in terms of various structural designs. With a reduced value of Young’s modulus, the 2024Al alloy exhibit a low wave impedance as compared to TC4. The tensile wave is reflected at the interface of TC4 and the 2024Al alloy and the much-reduced portion of it is transmitted to the 2024Al alloy. At a high strain rate, the 2024Al alloy shows a reduction in hardening; therefore, when the high stress induces plastic deformation, the plastic wave is transformed into a continual deviating form.

Elaborating it further, it can be said hypothetically that the plastic deformation in heterostructure composites results in the nonuniform deformation of distinct layers, which induce back stress in the soft layer and forward stress in the hard layer. This generates hetero-deformation-induced (HDI) strengthening in a composite structure, which ultimately increases yield strength and improves strain hardening, thus preserving ductility. This phenomenon is obvious in Figure 4a (at 1000/s strain rate). The back stress is long-range internal stress, which is generated due to dislocation pile-ups or geometrically necessary dislocations (GNDs). It potentially balances the applied stress, delays dislocation occurrence and slips in the soft layer, which increases its strength. The forward stress is generated in the hard layer due to a stress concentration at the interface, because of the collisions among GNDs. HDI (hetero-deformation-induced) hardening occurs as a result of a correlated constrained effect of forward and back stress in hard and soft zones, respectively. When it dominates the dislocation hardening, it can potentially enhance the ductility. Therefore, it is also called kinematic stress because there is stress redistribution away from the soft (plastically deforming) to hard (nondeforming) layer, which can generate uneven deformation paths, thereby increasing work hardening. The strain gradient is produced to accommodate the strain difference between the two layers and create a boundary-layer affected region. It is noteworthy that the forward and back stress at the boundary layer should be equal but opposite in direction, thus enhancing the strength of the layered composite structure without compromising ductility.

The increased strain hardening also increases hardness and wave impedance at 1000/s. At this time, there is a probability that the wave impedance gradient of two distinct material layers in a composite structure decrease and facilitate the wave transmission. If the wave impedance of the back layer is less than the front layer, the impact resistance can be accounted for in a constructive relation with the impact velocity. This correlation can be used to describe the energy diminution and supports, the phenomenon that, with a slow increase in strain, the strain rate hardening increases the energy diminution ability of the layered composite structure. The stress wave impedance is given by [29]
(6)$$C=\sqrt{\frac{E}{\rho }}$$
where *E* is Young’s modulus and *ρ* is the density of the material. In this heterostructure composite system, the wave impedance of both the materials are initially the same but, at a high strain rate, the strain rate hardening of TC4 is more than the 2024Al alloy, which in turn effects the wave impedance gradient and resultantly the energy diminution capability. At a much-increased strain rate of 6500/s (32.6 m/s striker velocity), the yield stress increased from 330 MPa at 1000/s to 467 MPa and the structure failure occurs at an increased strain, i.e., at 0.3; see Figure 4b. At this strain rate, the stress wave oscillations were smoothed and the elastic wave generated at the impact point was abruptly followed by a plastic wave within fractions of a second, thus propagating through TC4, resulting in a nominal thickness reduction and giving a maximum time to 2024Al for plastic deformation. It can be inferred that at room temperature and a 1000/s strain rate, the layered composite structure acted as a single entity and exhibited an enhanced impact resistance, as is obvious in Figure 7.

In the heterostructure composite system, there is an assumption that the dissimilar metals/alloys may join immediately after a high-speed impact and produce a diffusion layer. This indicates that when the shear stress wave reaches the interface layer, supposedly, the adiabatic temperature rise is very high and results in solid-state joining. The diffusion layer may comprise an intermetallic or metastable layer. If this layer is thin and strong, then it can improve the overall plastic deformation capacity by improving deformation harmonization [33]. This may occur due to instant heating and cooling during a high-speed impact. The difference in Young’s modulus and the thermal coefficient of the expansion of TC4 and 2024Al generated high residual compression and residual tensile stress in the layered composite structure; hence, the dominant residual stress is responsible for the strengthening and toughening mechanism in the layered composite structure. The deformation behavior of layered structures under a high-speed impact undergoes the same mechanism of gradient residual stresses. In case the crack initiation takes place in the compressive stressed layer, it will reduce the stress intensity factor and crack propagation velocity. Resultantly, the crack is diverged. But if the crack succeeds in reaching the tensile stressed layer, it facilitates intergranular cracks and relieves the localized stress concentration [34]. The same occurred in the case of the TC4-2024Al composite layered structure at a higher strain rate, which is obvious from the sample morphology in Figure 7.

## 4. Analytical Model of Layered Composite Material System

In this study, for both layers in the hetero-structured composite system as shown in Figure 8, two ductile materials with different properties, specifically hardness, were taken to analyze the deformation behavior at a high-speed impact. For translating the deformation mechanism, initially the modified Thomson model for the single-layer target and Florence model (based on areal density) were combined. These two models were correlated to translate controlling factors from the perspective of the armor protection of the heterostructure composite material system, and a modified Florence model in terms of energy absorption is proposed. Different factors were considered in establishing the mathematical model for the hetero-structured composite material system (both ductile layers) as a target plate, such as the target morphology, deformation of the projectile/eroded mass, material behavior on the first target layer on impact and material behavior on the second target layer on impact.

In the case of the heterostructure composite material system, the energy absorbed will be under:(7)$$E_{t}=E_{p}+E_{1}+E_{2}+E_{3}$$
here
(8)$$E_{p}=\frac{\rho_{b}h_{b}A_{b}v_{i}{}^{2}}{2}\;$$
(9)$$E_{1}=\;\pi r_{b}^{2}h_{\mathrm{fl}}\left(\frac{1}{2}Y_{\mathrm{fl}}+1.86\rho_{\mathrm{fl}}{\left(\frac{v_{i}r_{b}}{l_{n}}\right)}^{2}\right)$$
(10)$$E_{2}=\;\pi h_{\mathrm{fl}}w_{c}^{2}Y_{\mathrm{fl}}e^{-2aC_{\mathrm{de}}}\left(1+2aC_{\mathrm{de}}\right)\times 4{\left(1-\mu +\mu^{2}\right)}^{\frac{1}{2}}$$

Equations (9) and (10) were obtained from the modified Thomson model [35], where $E_{1}$ is the energy in the through-thickness direction of the front layer and $E_{2}$ is the energy that propagates in the radial direction in the front layer. When the leftover energy is sufficient enough to reach the second layer and travel through it, it can be given by $E_{3}$ based on the Florence model [36].
(11)$$E_{3}=\frac{1}{2}\;\left(\frac{\pi \beta_{2}x^{2}h_{\mathrm{bl}}\left(m_{\mathrm{br}}+\pi x^{2}\left(\rho_{\mathrm{fl}}h_{\mathrm{fl}}+\rho_{\mathrm{bl}}h_{\mathrm{bl}}\right)\right)}{m_{\mathrm{br}}}\right)$$

In order to obtain *E*_3_, consider that the plugged projectile is moving with a ballistic limit velocity after its encounter with the first layer and imparts the remaining energy to the composite layer behind. The residual kinetic energy that is absorbed is given by
(12)$$E_{3}=\frac{1}{2}m_{\mathrm{br}}v_{\mathrm{bl}}^{2}\;$$

As per the Florence model [36], *v*_bl_ is given by
(13)$$v_{\mathrm{bl}}^{2}=\frac{\pi \beta_{2}x^{2}h_{\mathrm{bl}}\left(m_{\mathrm{br}}+\pi x^{2}\left(\rho_{\mathrm{fl}}h_{\mathrm{fl}}+\rho_{\mathrm{bl}}h_{\mathrm{bl}}\right)\right)}{m_{\mathrm{br}}^{2}}$$
where
(14)$$\beta_{2}=\frac{\varepsilon_{2}\sigma_{2}}{0.91}$$
(15)$$x=r_{b}+2h_{\mathrm{fl}}$$

The areal density of the composite structure $\hat{A}$ will be
(16)$$\hat{A}=\rho_{\mathrm{fl}}h_{\mathrm{fl}}+\rho_{\mathrm{bl}}h_{\mathrm{bl}}$$

Now, the value of *v*_bl_ from Equation (13) is put in Equation (12):(17)$$E_{3}=\frac{1}{2}m_{\mathrm{br}}\left(\frac{\pi \beta_{2}x^{2}h_{\mathrm{bl}}\left(m_{\mathrm{br}}+\pi x^{2}\left(\rho_{\mathrm{fl}}h_{\mathrm{fl}}+\rho_{\mathrm{bl}}h_{\mathrm{bl}}\right)\right)}{m_{\mathrm{br}}^{2}}\right)$$
(18)$$E_{3}=\frac{1}{2}\;\left(\frac{\pi \beta_{2}x^{2}h_{\mathrm{bl}}\left(m_{\mathrm{br}}+\pi x^{2}\hat{A}\right)}{m_{\mathrm{br}}}\right)$$

The energy left to be absorbed by the back layer can potentially be calculated through Equation (18). The area undergone under deformation in this zone is given by *β.* Therefore, by putting all the respective energy values together, the total energy *E*_t_ is given by
(19)$$E_{t}=\frac{\rho_{b}h_{b}A_{b}v_{p}{}^{2}}{2}+\pi r_{b}^{2}h_{\mathrm{fl}}\left(\frac{1}{2}Y_{\mathrm{fl}}+1.86\rho_{\mathrm{fl}}{\left(\frac{v_{i}r_{b}}{l_{n}}\right)}^{2}\right)+\frac{1}{2}\;\left(\frac{\pi \beta_{2}x^{2}h_{\mathrm{bl}}\left(m_{\mathrm{br}}+\pi x^{2}\hat{A}\right)}{m_{\mathrm{br}}}\right)$$

Whereas the law of conservation of energy of a whole system is given by
(20)$$\frac{1}{2}mv_{i}^{2}=\frac{1}{2}mv_{r}^{2}+E_{t}\;$$

The above equation shows the energy balance of a system, thereby considering the absorbed energy and energy associated with the residual velocity, which may be calculated separately. In this case, the kinetic energy transferred to the system was calculated as 15.2 J, *E*_t_ was calculated through the proposed analytical model—which is also 15.2 J—and residual velocity approached zero with time; the first term of Equation (19) will become 0 J. Hence, it is seen that the energy was conserved in this system, which also indicates the elastic mode of collision. In addition, the performance of these heterostructure composites also depends upon the thickness of each layer relative to each other. Therefore, in order to calculate the thickness ratio of the target plate, *λ* was introduced [37], and it ranges from 0.6 to 1.6 according to literature studies. It is given by
(21)$$\lambda =\frac{h_{\mathrm{fl}}}{h_{\mathrm{bl}}}$$

In case of a multilayer target, the relative thickness of each layer with respect to its above layer is considered and the above equation can be re-written as
(22)$$\lambda =\sum (h_{1}:h_{2}:h_{3}:\ldots \ldots \ldots :h_{i})$$

Putting the value of $h_{\mathrm{fl}}$ from Equation (15) in Equation (21), we obtain
(23)$$\lambda =\frac{\left(x-r_{b}\right)}{2h_{\mathrm{bl}}}$$

From this equation, the minimum required thickness ratio can be obtained. This model can be used for multilayers as well. For instance, in the case of three layers, the above equation will be
(24)$$\lambda =\frac{h_{\mathrm{fl}}}{h_{\mathrm{bl}1}+h_{\mathrm{bl}2}}$$
and, after adding the value of *h*_fl_, we will obtain
(25)$$\lambda =\frac{\left(x-r_{b}\right)}{2(h_{\mathrm{bl}1}+h_{\mathrm{bl}2})}$$

The aforementioned parameter enables the determination of the relative thicknesses in a heterostructure composite material system. Hence, in view of the above, the front layer of TC4 was taken as 5 mm and the backup layer of the 2024Al alloy as 7 mm, so λ for this structure is 0.7, which falls within the set limit.

## 5. Numerical Simulation of Projectile Target Impact of Layered Structure

### 5.1. Numerical Model

The material model used for the proposed layered heterostructure composite material system is shown in Figure 9. The material model used in the impact analysis was Mat-modified Johnson Cook (107) for the target composite structure, and tabulated parameters in Table 2 are for the TC4 alloy and Table 3 are for the 2024Al alloy, while the Johnson Cook model (015) was used for the bullet, as shown in Table 4. It is worth mentioning here that material parameters were taken from the literature. The meshing of the model was performed by following general rules for FEA. The total number of nodes utilized for calculations was 122,807 and the number of elements was 110,224. The contact option “ERODING_SURFACE_TO_SURFACE” was used, while EOS_GRUNEISEN was used to describe the hydrodynamic pressure involved during the high-velocity impact. The time for the impact analysis was set to be 120 µs, whereas the velocity of the impacted projectile is 700 m/s, and the mass of the bullet is 6.25 g. Furthermore, in order to obtain an impact effect close to the real 7.62 mm bullet impact, the material model of the bullet was optimized to obtain precise simulation results.

The constitutive modelling and numerical simulations together can potentially provide the base to propose the heterostructure composite (functionally gradient layered structure) material system as protective armor. For the design performance of protective armors, the areal density is the most fundamental parameter, especially when a light weight is required. As mentioned earlier, the lightweight layered heterostructure composite material system comprises TC4 as the front layer (5 mm thick) and the 2024Al alloy as the back layer (7 mm thick), and their combined areal density becomes 40.9 kg/m^2^, which is an approximately 43% weight reduction as compared to armor steel of the same thickness. This gives a wide bracket of difference that can be utilized for further design optimization as well. Being lightweight and frequently used alloys in the aircraft industry, these two materials (TC4, 2024Al) were combined to obtain their benefits in an improved way. TC4 possess an excellent energy absorption capability, high strength, less density and high hardness. Due to these advantages, it is the leading candidate among metallic materials used as a front or backing layer in a heterostructure composite material system [29,41], whereas 2024Al is also extensively used in the aircraft and aerospace industry due to its high strength to weight ratio and good fatigue resistance properties [42].

### 5.2. Numerical Results

(a)Stress distribution analysis

When a projectile is impacted on this layered heterostructure composite target, there is an abrupt stress rise until 1421 MPa, then it attains a steady state at approximately 1471 MPa, followed by a sudden drop in stress down to 692 MPa. After this point, there is an immediate recovery jump in stress back to 1480 MPa and it attains a steady state onwards until *ε* = 0.14, after which there is a rapid drop in stress. This was the case of the stress response in the front layer, whereas in the back layer, the stress was much less, and after attaining the peak stress of approximately 500 MPa, it becomes steady. Furthermore, there was more stress and strain experienced by the front layer as compared to the back layer; see Figure 10.

It can be seen that, initially, the stress is concentrated in the center of the impacted plate until it reaches its maximum value at 28 µs, and then it spreads outward throughout the two layers. Finally, it pushes the projectile just like a bouncing ball, thus exhibiting the elastic collision behavior; see Figure 11. The impact mechanism can be well visualized through Figure 12. The stress distribution pattern is different for the two plates and is obvious in Figure 13. It may be used to analyze the impact compression behavior of a material [30]. When compressed slowly, the composite heterostructure obtains sufficient time to counter the effect of the applied load in the form of multiple deformation modes over large strain values and this is obvious from the prolonged elastic region followed by the plastic region, leading to the rupture of material. This also facilitates the maximum load shift from the front plate towards the back plate, thus making the front plate a compression tool and all of the incoming load is transferred to the second plate. Here, the two plates act as different entities and their reaction to the applied load is different. Different layers exhibit a different stress wave propagation, which is the key factor that can be tailored to obtain a required anti-penetration performance from this type of composite structures. It also makes the selection of materials more appropriate with respect to a requirement. As the front layer is the TC4 alloy, the stress wave propagation is initially star-shaped, which is converted into a circular wave propagation, and stress concentration points are in exact opposite directions with respect to each other and evenly distributed throughout the plate cross-section. Due to the efficient stress wave propagation in the front layer, a very small amount of stress is transferred towards the back layer. In contrast, the back layer, made up of the 2024Al alloy, exhibited a totally different wave propagation pattern. Initially, it takes a square shape at maximum stress, which is then changed into a quad-shaped pattern extending towards the plate corners.

It can therefore be inferred from this pattern that the combined performance of the heterostructure composite (TC4, front hard layer; 2024 Al, back soft layer) is strong against a high-speed impact. Finally, after 120 µs, the stress concentration points are confined towards the corner with minimum stress in the center. Overall, the opposite directional stress wave propagation delays stress concentration and strain localization [43], thus enabling the heterostructure composite to show better anti-penetration behavior against the incoming threat.

Similarly, the velocity time graph in Figure 14a shows that the plug formation starts at 10 µs and after that there is a sharp slopy decline in velocity until 44 µs, which is the point of the maximum adiabatic temperature rise. There is again an abrupt rise in stress after falling, i.e., from 692 MPa to 1480 MPa in Figure 10. This also shows that there is a maximum transition in stress and its distribution throughout the cross-section of the two layers. After this time period, there is slow change in the velocity and the parameters correlated with it. The curvy slope of energy absorption with time is the same (Figure 14b), which shows that energy and velocity are directly proportional to each other. After the major stress distribution in the front layer, the minor stress is transferred towards the back layer (although less in value), yet it regulated stress wave propagation while enhancing the anti-penetration capability. It is also worth mentioning that the stress wave propagation along the circumferential/radial direction is more intensive than the through-thickness direction that can be considered as one of the advantages of the heterostructure composite material system.

(b) Adiabatic temperature analysis

The simulation results from the perspective of adiabatic temperature also point out the fact that two materials with different thermal properties reduced the adiabatic temperature rise and also gave rise to a temperature gradient. The illustrative results for the adiabatic temperature rise from the front view are shown in Figure 15, and after 44 µs, there is notably no change in the adiabatic temperature value. It can be seen that at 20 µs, the adiabatic temperature rise tends to reach the back layer and is confined to the impact point only in a very close vicinity. The maximum temperature rise is up to 365.8 K at 44 µs in the front layer, while in the back layer, it is reduced to 304 K, as shown in Figure 16. Here, different materials in the layered composite structure also facilitated the reduction in the adiabatic temperature rise, which means that there is less deformation damage corresponding to a temperature influence. The temperature gradient regulation between the two plates reduced the heat dissipation that can be used in the plastic work, thereby reducing the eventual deformation damage induced in the layered structure. This in turn also imparts its effect on the microstructures in this region. The effect of adiabatic temperature rise combined with corresponding stress controls the grain size and its distribution in these areas and has been discussed widely in the literature [44]. This is the key factor that is responsible for an improvement in ductility as well as toughness of a material. In Figure 16, there is a steady state in temperature after an initial abrupt rise until 10 µs and it continues for the next 15 µs; the same steady state can be observed in Figure 15c,d. This time period also has a steep decline in velocity and energy, which indicates that during this time span, the energy transfer to the specimen in the form of heat dissipation is large and the same has been transferred to the back layer but at a constant rate. It raised the temperature of the back plate as well, and after this, the energy transfer in the form of heat increased and the temperature of the back plate started rising significantly.

### 5.3. Comparative Analysis

As discussed in a previous section that the composite structure is comprised of two ductile materials (TC4, a hard front layer, and the 2024Al alloy, a comparatively soft back layer) and the effect of interlayer formation at the joint is not considered. Upon a high speed, the two layers responded to the incoming threat as a combined entity and exhibited an elastic collision. By using the analytical model (improved Florence model), the energy absorption of the system was calculated, which was validated through numerical simulation results. They are appended below in Table 5.

The difference in energy transfer *E*_3_ may result due to the adiabatic temperature rise effect. The analytical method measures the energy absorption capability based on the thickness of the target whereas, in the numerical simulation, the effect of temperature during impact cannot be ruled out. Therefore, this fluctuation in energy values is a result of the temperature rise and associated parameters that are generated in the back plate after impact. It can be seen that both the analytical model and simulation model values with their segregation are in good agreement. In addition, the reduced yield stress of the front layer and increased yield stress of the back layer, when responding in a combined way under the influence of hetero-deformation-induced (HDI) strengthening, exhibited an improved ductility and strength simultaneously, which is contrary to the general strength–ductility tradeoff concept. This in turn enhanced the impact resistance ability of the proposed layered heterostructure composite material system. It has also been observed that the stress was distributed in the front and back layer with an outward direction, thereby making its exit through the edges of the structure. Similarly, the adiabatic temperature rise for the two plates showed a temperature gradient of 61 K. This may impart its effect on the microstructural change specifically in the back plate where there was an increase in stress but the temperature rise was not significant, which means that this is a stressed region, but there is no thermal softening effect observed and this resulted in a fine-grained microstructure. In this region, there is a considerable increase in the proportion of recrystallized grains and they are evenly distributed. This corresponds to the fact that dDRX (dynamic recrystallization) was initialized during the deformation; it eliminated the dislocations and refined the grain size, thereby enhancing the toughness and ductility. Therefore, it can be inferred that the result of a high stress and low thermal softening generates uniformly distributed equiaxed fine grains, as reported in various studies [28,35].

## 6. Conclusions

A study was undertaken to reveal the design and performance of an emerging class in protective armors, layered heterostructure composites (proposed structure: front TC4 layer backed by 2024Al layer), in the light of their deformation behavior on high-speed impacts. The enhanced absorption capability and anti-penetration behavior was investigated and the most widely accepted Florence model was used. In this regard, some modifications were proposed for the model, so that it can be utilized for heterostructure composite structures with a high precision. Numerical simulation was performed by using Ls-Dyna code to validate the analytical model findings. The following are the conclusions:a.The proposed layered heterostructure composite system was tested with the SHPB test. The results revealed that at a high-speed impact, the layered heterostructure composite structure system exhibited an enhanced impact resistance under the influence of hetero-deformation-induced (HDI) hardening and the combined effect of back stress and forward stress in the layers of the composite structure. The structure behaved as a unified entity to offer resistance to the high-speed impact.b.The strain rate sensitivity m decreased with an increase in the strain rate while the strain energy density $\hat{U}$ exhibited a significant increase when the strain exceeded 0.14, and at lower strain values, the strain energy density was almost similar, showing no response to an increased strain rate.c.The proposed modifications in the Florence model to translate the energy absorption and target the plate thickness revealed that the heterostructure composite material system has energy distribution in such a way that the maximum energy is absorbed by the ductile and hard front layer and a much lesser amount is transferred to the soft ductile backing layer.d.When the two ductile materials with different properties and layer thicknesses were joined together, they responded to the incoming projectile like an elastic collision. The heterostructure composite material system exhibited a slight bend and bounced back the projectile mass probably due to the fact that the yield strength of the front layer was decreased while the back layer’s strength was increased. Their collective response enabled the composite structure to show an enhanced ductility with strength.e.The numerical simulation of the proposed heterostructure composite validated the findings of the analytical model, and the energy distribution values are in good agreement with each other. Hence, the proposed heterostructure composite material system can potentially be utilized as a protective armor system.

## Figures and Tables

**Figure 1 materials-16-05169-f001:**
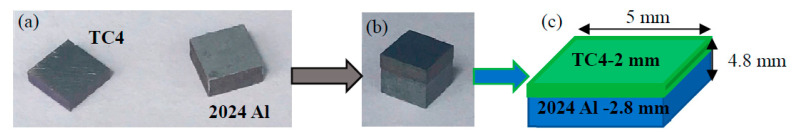
SHPB sample: (**a**) separated alloy specimen, (**b**) joined layered heterostructure through chemical adhesive and (**c**) sample dimensions.

**Figure 2 materials-16-05169-f002:**
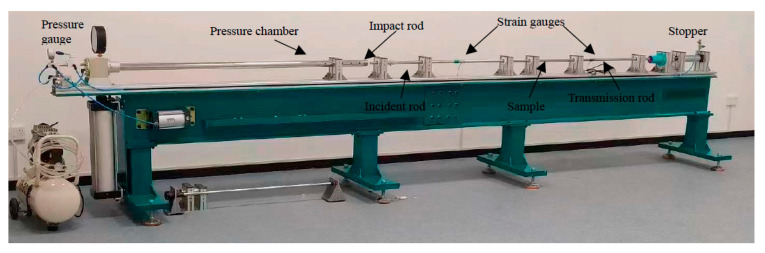
SHPB experimental apparatus.

**Figure 3 materials-16-05169-f003:**
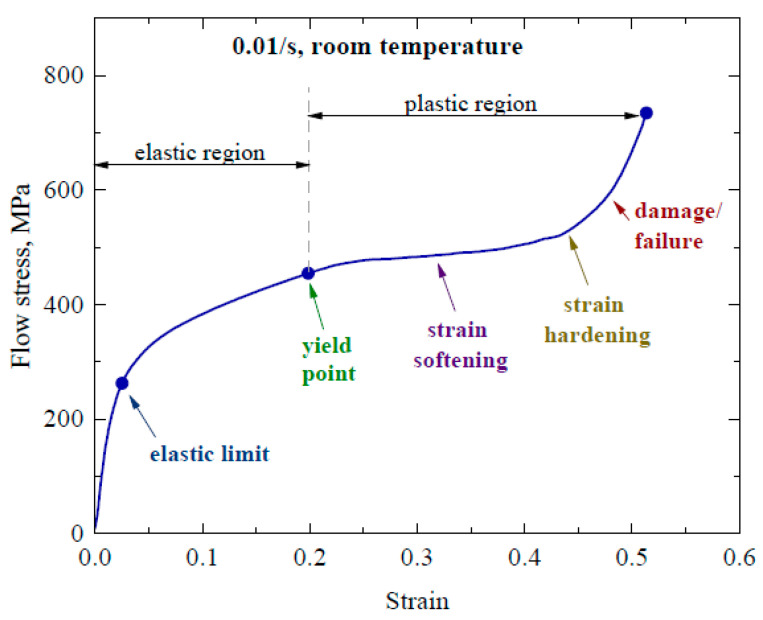
Stress–strain curve at 0.01/s (quasistatic condition and room temperature).

**Figure 4 materials-16-05169-f004:**
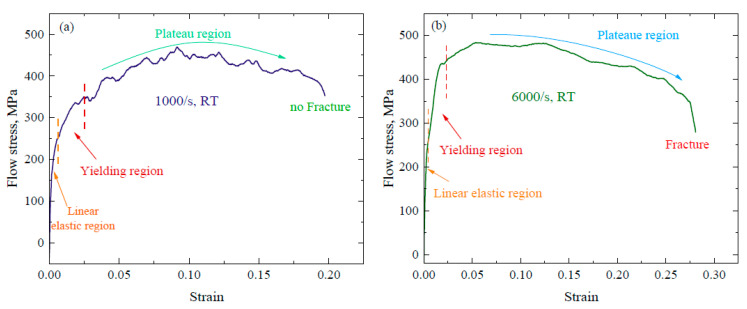
Stress–strain curve at room temperature: (**a**) 1000/s strain rate and (**b**) 6500/s strain rate.

**Figure 5 materials-16-05169-f005:**
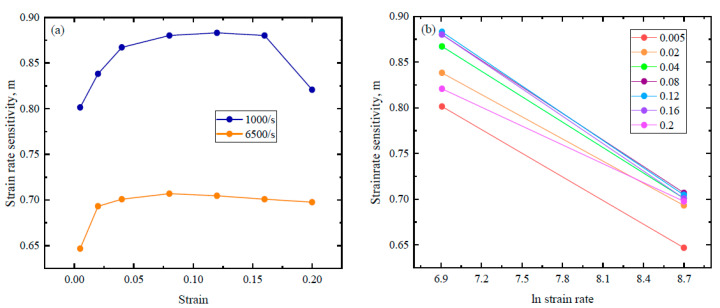
Strain rate sensitivity vs: (**a**) strain and (**b**) strain rate.

**Figure 6 materials-16-05169-f006:**
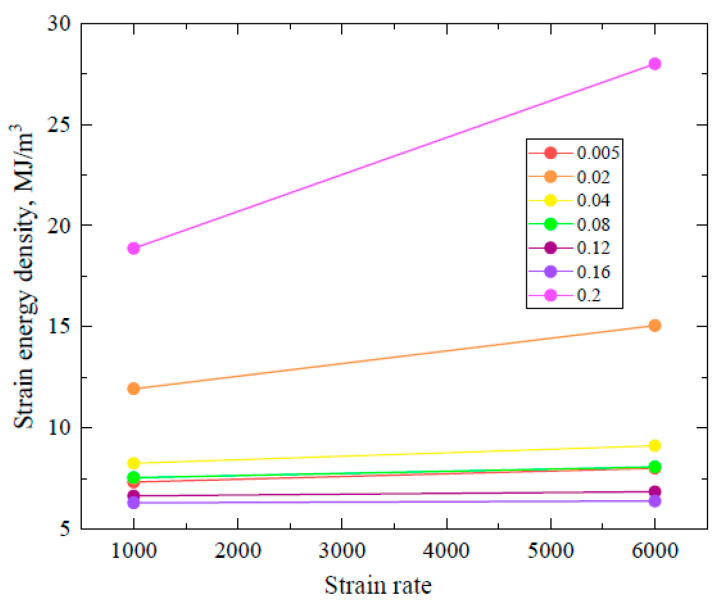
Strain energy density vs. strain rate at different strain values.

**Figure 7 materials-16-05169-f007:**
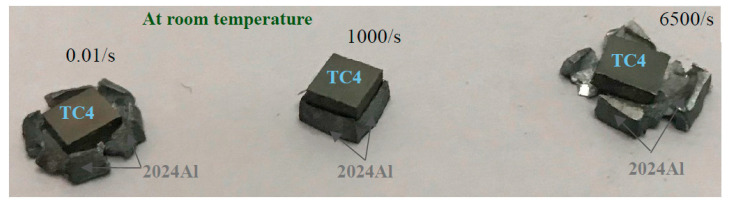
Macro features of the layered composite structure (TC4 + 2024Al) at room temperature and different strain rates.

**Figure 8 materials-16-05169-f008:**
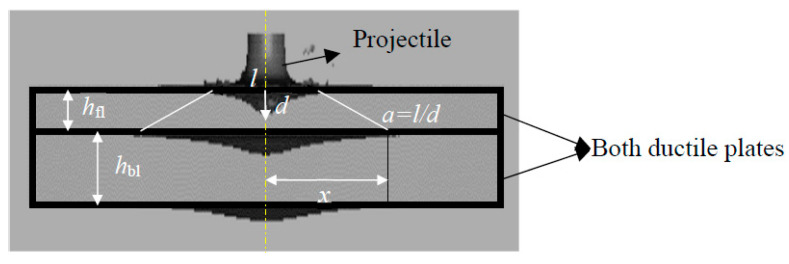
Numerical model of the proposed composite structure.

**Figure 9 materials-16-05169-f009:**
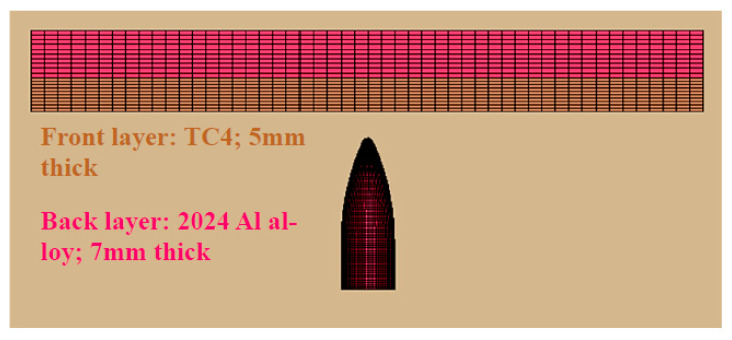
Numerical model of the proposed layered heterostructure composite system.

**Figure 10 materials-16-05169-f010:**
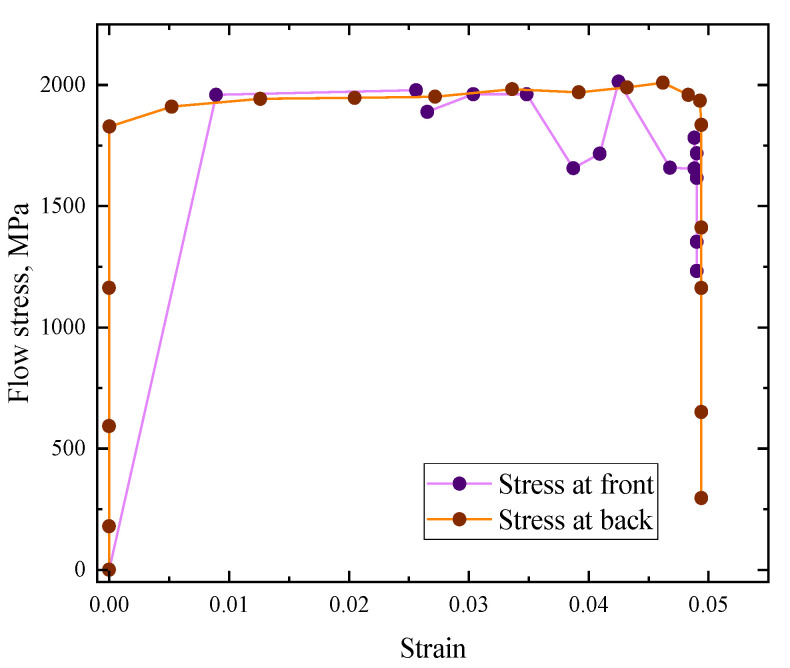
Stress–strain curve for the front and back layer.

**Figure 11 materials-16-05169-f011:**
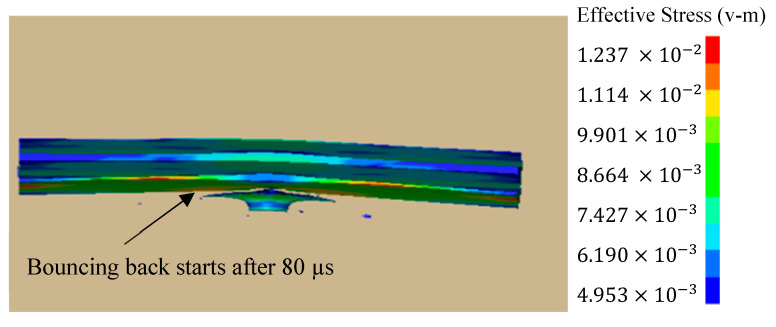
Bouncing-back effect of the projectile.

**Figure 12 materials-16-05169-f012:**
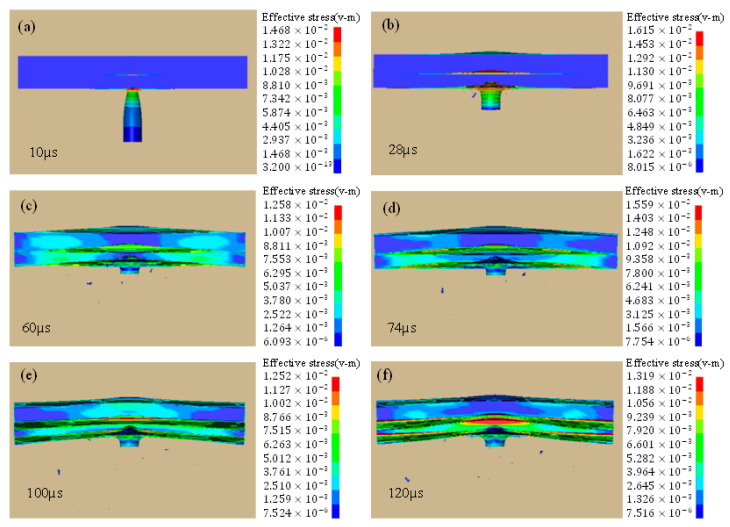
High-speed impact showing behavior of the front-layer and back-layer response at different time (**a**) 10 µs, (**b**) 28 µs, (**c**) 60 µs, (**d**) 74 µs, (**e**) 100 µs and (**f**) 120 µs.

**Figure 13 materials-16-05169-f013:**
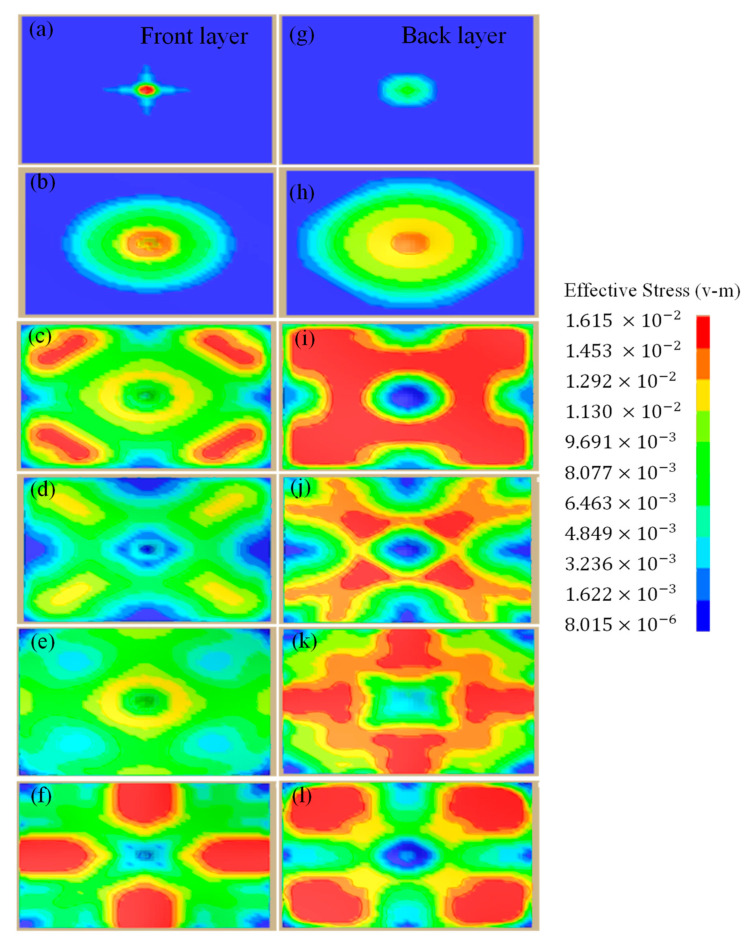
Stress distribution on the front and back layer at (**a**,**g**) 10 µs, (**b**,**h**) 28 µs, (**c**,**i**) 60 µs, (**d**,**j**) 74 µs, (**e**,**k**) 100 µs and (**f**,**l**) 120 µs, respectively.

**Figure 14 materials-16-05169-f014:**
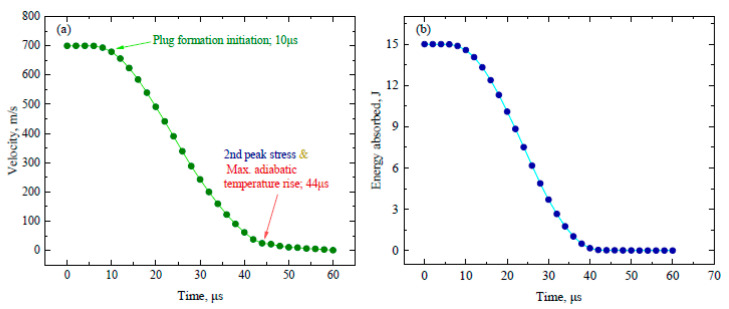
Graphs of (**a**) velocity vs. time and (**b**) energy absorption vs. time.

**Figure 15 materials-16-05169-f015:**
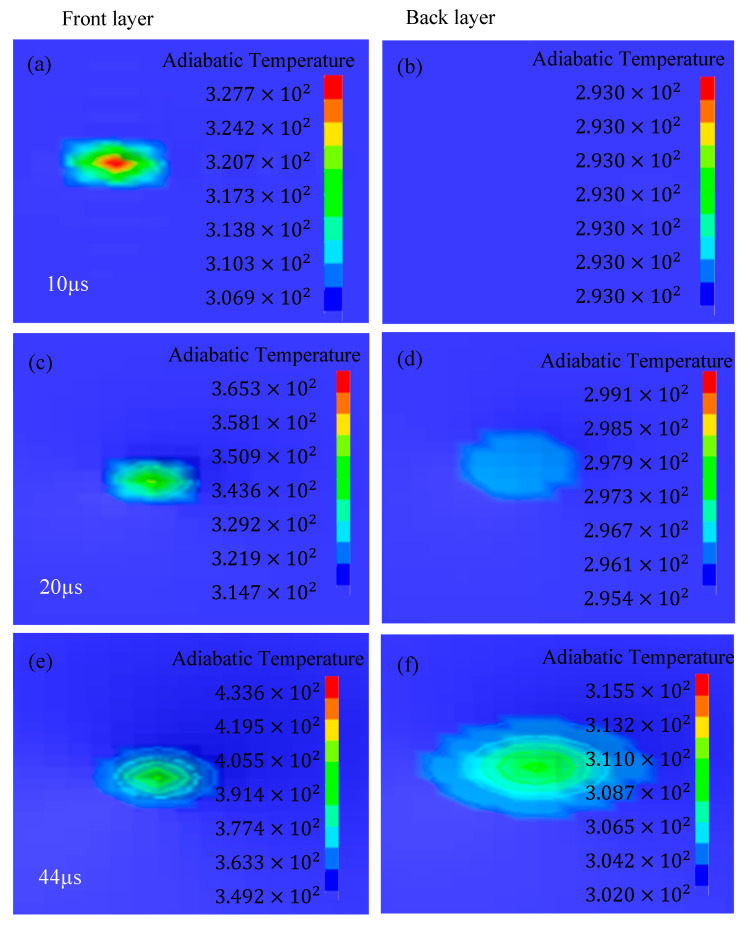
Adiabatic temperature (history variable #4) distribution on the front and back layer at (**a**,**b**) 10 µs, (**c**,**d**) 20 µs and (**e**,**f**) 44 µs, simultaneously.

**Figure 16 materials-16-05169-f016:**
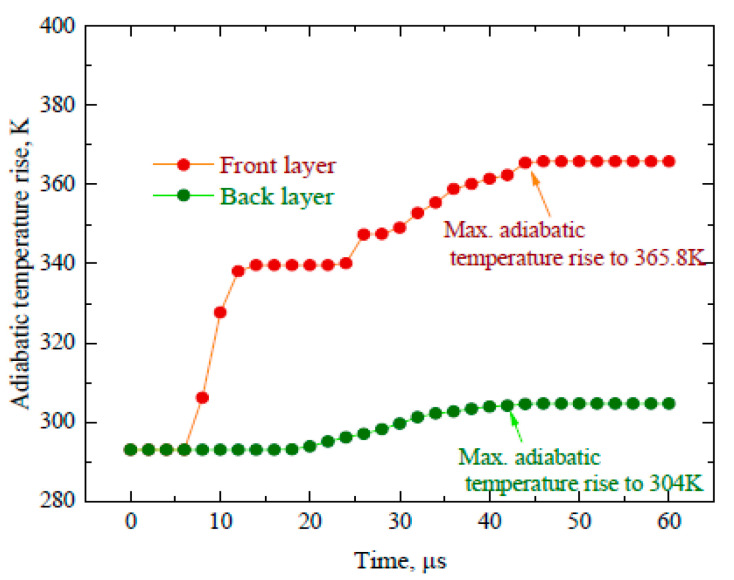
Adiabatic temperature rise vs. time.

**Table 1 materials-16-05169-t001:** Schematics of quasistatic (Instron 3382 UTS) and dynamic (SHPB) tests.

Sample Qty Tested at Room Temperature	Strain Rate (/s)
03	0.01
03	10^3^
03	6.5 × 10^3^

**Table 2 materials-16-05169-t002:** Modified Johnson Cook (MJC) model parameters for target material, TC4 [35,38,39].

Material Property	Symbol	Unit	TC4
Density	RO	g/cm^3^	4.42
Young’s modulus	E	GPa	113
Poisson ratio	PR		0.33
Specific heat	CP	J-kg/K	580
	A	GPa	1.089
	B	GPa	1.083
	m		1.1
	C		0.014
	n		0.93
Fracture parameters	D_1_		−0.09
	D_2_		0.27
	D_3_		0.48
	D_4_		0.014
	D_5_		3.86

**Table 3 materials-16-05169-t003:** Modified Johnson Cook (MJC) model parameters for target material, 2024Al alloy [35,40].

Material Property	Symbol	Unit	2024Al Alloy
Density	RO	g/cm^3^	2.78
Young’s modulus	E	GPa	71
Poisson ratio	PR		0.31
Specific heat	CP	J-kg/K	236
	A	GPa	0.290
	B	GPa	0.456
	m		1.0
	C		0.0037
	n		0.301
Fracture parameters	D_1_		0.034
	D_2_		0.664
	D_3_		−1.5
	D_4_		0.011
	D_5_		0

**Table 4 materials-16-05169-t004:** Johnson Cook (JC) model parameters for a steel projectile [35].

Material Property	Symbol	Unit	Hard Steel Core
Density	RO	g/cm^3^	7.8
Young’s modulus	E	GPa	200
Poisson ratio	PR		0.28
Specific heat	CP	J-kg/K	450
	A	GPa	0.49
	B	GPa	0.807
	m		0.94
	C		0.0108
	n		0.73
Fracture parameters	D1		1.2
	D2		0.732
	D3		−0.24
	D4		−0.015
	D5		0
EOS parameters	S1		1.49
	γ		2.17
	A		0.46

**Table 5 materials-16-05169-t005:** Comparative analysis of absorbed energy *E*_t_ with analytical and simulation models.

	*E*_p_ (J)	*E*_1_ (J)	*E*_2_ (J)	*E*_3_ (J)	*E*_t_ (J)
Analytical model	3.6	8.0	1.5	2.1	15.2
Simulation model	3.0	7.2	1.4	3.4	15.0

## Data Availability

Data will be made available upon request.

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
