# Peer review of "Design and Performance of Layered Heterostructure Composite Material System for Protective Armors"

_materials, 2023, doi:10.3390/ma16145169_

Round 1

Reviewer 1 Report

This paper presents.

A new layered heterostructure composite material system (TC4 as front layer and 2024Al alloy as back layer) has been developed and analyzed for their design and performance in terms of enhanced absorption capability and anti-penetration behavior. The Florence model for energy absorption has been modified, so that it can be utilized for layered heterostructure composite material system with more efficacy. Numerical simulation through Ls-Dyna validated the analytical model findings regarding energy absorption of the system and both were in good agreement. Results have shown that two ductile materials with diverse properties specifically, hardness gradient and varied layer thickness joined together, behaved like a unified structure and exhibited elastic collision after slight bending, which is possibly due to the decreased yield strength of front layer and increased yield strength of second layer. To validate the analytical and numerical findings the samples of layered heterostructure composite material system has been subjected to SHPB (split Hopkinson pressure bar) compression test. The deformation behavior has been analyzed in context of strain energy density and stain rate sensitive parameter at different strain rates. The encouraging results proposed that two ductile materials with hardness gradient can be used as an alternate structure instead of brittle-ductile combination in a layered structure.

This paper is written well and deserves for possible publication in this journal. The following minor issues need to be addressed before publication.

1.     Figures 11, 13, and 15 quality is not good revise them with at least 300 dpi

2.     Add the following related update to the old references and improve the quality of the paper.

- Influence of elbow angle on erosion-corrosion of 1018 steel for gas–liquid–solid three-phase flow

-Synthesis, surface nitriding and characterization of Ti-Nb modified 316L stainless steel alloy using powder metallurgy

Author Response

1) Figures 11, 13, and 15 quality is not good revise them with at least 300 dpi.

Response: Figures 11, 13 and 15 have been revised as per suggestion of the reviewer.

2) Add the following related update to the old references and improve the quality of the paper.

- Influence of elbow angle on erosion-corrosion of 1018 steel for gas–liquid–solid three-phase flow

-Synthesis, surface nitriding and characterization of Ti-Nb modified 316L stainless steel alloy using powder metallurgy

Response: The references have been added to the manuscript in introduction section and highlighted yellow.

Reviewer 2 Report

In case of adhesive bonding the influence of the adhesive can not be neglected (See studies on combined joining processes like clinching/adhesive bonding or blind riveting/adhesive bonding. I expect severe debonding in case of an impact.

The strain rate dependency evaluation is kind of strange: SHPB for low strain rates? A combination with other test methods for strain rates up to 100/s would be suitable!

The simulation results are not over all consistent with literature. Especially the adiabatic heating differs quite a lot!

The image quality (scaling, resolution, colors) should be improved especially for the numerical simulation.

Introduction parts are also included in the later chapters again.

Vocabulary is OK sometimes it seems to be a computer translation - unusual formulations are used.

The gramma should be improved significantly. Very long sentences unclearly linked together.  Some paragraphs are really hard to understand at all.

Author Response

1) In case of adhesive bonding the influence of the adhesive cannot be neglected (See studies on combined joining processes like clinching/adhesive bonding or blind riveting/adhesive bonding. I expect severe debonding in case of an impact.

Response: It is submitted that adhesive bonding has been used in this study to obtain excellent joint at room temperature however the role of adhesive bonding in joining process is beyond the scope of current study. The results in case of impact has been illustrated in Fig. 7. In the future, for industrial applications, explosive welding technology can be used and can form an ideal interface.

2) The strain rate dependency evaluation is kind of strange: SHPB for low strain rates? A combination with other test methods for strain rates up to 100/s would be suitable!

Response: The quasi-static compression test at strain rate of 0.01/s was conducted on Instron 3382 universal testing machine as per ASTM standard while dynamic compression test was carried out on Split Hopkinson Pressure Bar (SHPB)  at strain rate 103/s and 6.5 x103/s and room temperature. This information has been included in manuscript and highlighted yellow for identification. 

3) The simulation results are not over all consistent with literature. Especially the adiabatic heating differs quite a lot!

Response: The simulation results have been obtained for a composite structure which exhibits the combined effect of two different materials joined together under specific impact conditions and cannot be compared due to its different operating environment.

4) The image quality (scaling, resolution, colors) should be improved especially for the numerical simulation.

Response: The observation made has been addressed and highlighted yellow in main manuscript.

5) Introduction parts are also included in the later chapters again.

Response: As per valuable suggestion of the reviewer the first paragraph from section 4 has been moved to the introduction section and highlighted yellow.

Reviewer 3 Report

The authors developed a new layered heterostructure composite material system TC4 and analyzed for their design and performance in terms of enhanced absorption capability and anti-penetration behavior. I think this topic is original and relevant in the field. They discussed considering with other published articles. The conclusions consistent with the evidence and arguments presented in this article. The references are appropriate.

1. Figure 10. : There is a negative peak in 0.09 strain. Explain this cause.

2. L469 : What happens when you compress it slowly, not impulsively ?

3. Fig. 16 : Explain the plane of flat layer between 10 micro sec and 25 micro sec.

4. Table 5 : The difference of E3 is larger than that of other energy. Explain the reason of this difference of E3.

1. L515 : “... is reduced to 304K, Fig. 16.”--->... is reduced to 304K as shown in Fig. 16.

2. Leave the space for one character during a number and the unit.

For example, 455MPa ---> 455 MPa,  304K ---> 304 K.

Author Response

1) Figure 10. : There is a negative peak in 0.09 strain. Explain this cause.

Response: The negative peak in 0.09 strain might be caused by automatic coordinates during simulated value drawing, and that has been checked again. The revised graph in Figure 10 according to the corrected values have been highlighted yellow.

2) L469 : What happens when you compress it slowly, not impulsively ?

Response: When compressed slowly the composite heterostructure gets sufficient time to counter the effect of applied load in the form of multiple deformation modes over large strain values and this is obvious from the prolonged elastic region followed by plastic region leading to rupture of material. This also facilitates the maximum load shift from front plate towards back plate thus making front plate a compression tool and transfer all coming load to second plate. Here the two plates act as different entities and their reaction to applied load is different.

3) Fig. 16 : Explain the plane of flat layer between 10 micro sec and 25 micro sec.

Response: In Fig 16 there is a steady state in temperature after initial abrupt rise till 10 µs and continues for next 15 µs, the same steady state can be observed in Fig. 15 (c & d). This time period has also steep decline in velocity and energy which indicates that during this time span the energy transfer to the specimen in the form of heat dissipation is large and the same has been transferred to the back layer but at constant rate. It raised the temperature of back plate as well and after this the energy transfer in the form of heat increased and the temperature of back plate started rising significantly.

4) Table 5 : The difference of E3 is larger than that of other energy. Explain the reason of this difference of E3.

Response: The difference in energy transfer E3 may result due to adiabatic temperature rise effect. The analytical method measures the energy absorption capability based on the thickness of target whereas, in numerical simulation the effect of temperature during impact cannot be ruled out. Therefore, this fluctuation in energy values is a result of temperature rise and associated parameters that are generated in back plate after impact.

 5) L515 : “... is reduced to 304K, Fig. 16.”--->... is reduced to 304K as shown in Fig. 16.

 Response:  The correction has been made and highlighted yellow.

6) Leave the space for one character during a number and the unit. For example, 455MPa ---> 455 MPa,  304K ---> 304 K.

 Response:  The correction has been made accordingly and highlighted yellow for identification.

Reviewer 4 Report

The manuscript "Design and performance of layered heterostructure composite material system for protective armors" submitted for publication on Materials has been reviewed. It deals with an emerging class in protective armors based on composite heterostructure made with TC4 (front) and Al 2024 (back). The structure has been tested by means of SHPB test at high-speed impact. The structure behaved as unified entity and the strain rate sensitivity "m" decreased at increased strain rate while strain energy density was almost similar.

The manuscript is novel and interesting. The structure is well-arranged and organized. English almost fine. In my opinion it can be accepted after the following minor revisions:

1) Double space (line 18, line 240 and line 329), please remove;

2) check apex ( cm3, line 119 and 124);

3) Ref. [41] is not called in the main text;

4) A Discussion section should be added, in order to separate results from discussion;

5) Kinetic conditions of projectile (impact speed, energy, mass etc.) are not reported in the model, please 

6) It is not clear how the interface between TC4 and Al is managed (line 134, chemical adhesive), please add and explain.

7) Ref. 13 is not complete. Please check.

Acceptable

Author Response

1) Double space (line 18, line 240 and line 329), please remove.

Response: Double space has been removed and highlighted yellow in the revised manuscript.

2) Check apex ( cm3, line 119 and 124).

Response:  The apex has been checked and corrected accordingly. It is also highlighted yellow for identification.

3) Ref. [41] is not called in the main text.

Response: Ref [41] now Ref [28] in revised manuscript has been called in the Result and discussion section and highlighted yellow for identification. It is used in comparative analysis section to support the possible phenomenon involved in improving the ductility and toughness of the heterostructure composite material system.

4) A Discussion section should be added, in order to separate results from discussion.

Response: Result and analysis of layered composite material system in section 3 (title) has been changed to Results and discussion for better understanding of readers. Same has been highlighted yellow.

5) Kinetic conditions of projectile (impact speed, energy, mass etc.) are not reported in the model, please.

Response: The kinetic conditions of projectile has been incorporated as per valuable suggestion of the reviewer in sec 5.1 (Numerical model) and highlighted yellow.  

6) It is not clear how the interface between TC4 and Al is managed (line 134, chemical adhesive), please add and explain.

Response: As mentioned earlier, the interfacial layers potentially generate intermetallics which are known to weaken the joints therefore morphology of interfacial layer is of vital importance. Foregoing in view of above and the structural requirements of proposed heterostructure composite material system the formation of intermetallics is not desirable. Hence thin samples from TC4 and 2024Al alloy were cut as per specified dimensions and were joined together by using commercially available chemical adhesive. This arrangement avoided the formation of inter-metallics and their adverse effect on the joint performance of the structure. Future oriented industrial applications are expected to adopt explosive welding methods.

7) Ref. 13 is not complete. Please check.

 Response:  Ref [13] now Ref [15] in revised manuscript has been checked, completed and highlighted yellow.